# Engineering polar vortex from topologically trivial domain architecture

Congbing Tan [1,2,3,17], Yongqi Dong [4,5,17], Yuanwei Sun [6,7,17], Chang Liu [8,9,17], Pan Chen [10], Xiangli Zhong [2], Ruixue Zhu [6,7], Mingwei Liu [1], Jingmin Zhang [7], Jinbin Wang [2], Kaihui Liu [7,11], Xuedong Bai [10], Dapeng Yu [11,12,13], Xiaoping Ouyang [2], Jie Wang [8,14✉], Peng Gao [6,7,11,15✉], Zhenlin Luo [5✉] & Jiangyu Li [3,4,16✉]

Topologically nontrivial polar structures are not only attractive for high-density data storage, but also for ultralow power microelectronics thanks to their exotic negative capacitance. The vast majority of polar structures emerging naturally in ferroelectrics, however, are topologically trivial, and there are enormous interests in artificially engineered polar structures possessing nontrivial topology. Here we demonstrate reconstruction of topologically trivial strip-like domain architecture into arrays of polar vortex in $(PbTiO_3)_{10}/(SrTiO_3)_{10}$ superlattice, accomplished by fabricating a cross-sectional lamella from the superlattice film. Using a combination of techniques for polarization mapping, atomic imaging, and three-dimensional structure visualization supported by phase field simulations, we reveal that the reconstruction relieves biaxial epitaxial strain in thin film into a uniaxial one in lamella, changing the subtle electrostatic and elastostatic energetics and providing the driving force for the polar vortex formation. The work establishes a realistic strategy for engineering polar topologies in otherwise ordinary ferroelectric superlattices.

[1] Hunan Provincial Key Laboratory of Intelligent Sensors and Advanced Sensor Materials, School of Physics and Electronics, Hunan University of Science and Technology, Xiangtan, Hunan, China. [2] School of Materials Science and Engineering, Xiangtan University, Xiangtan, Hunan, China. [3] Department of Materials Science and Engineering, Southern University of Science and Technology, Shenzhen, Guangdong, China. [4] Shenzhen Key Laboratory of Nanobiomechanics, Shenzhen Institutes of Advanced Technology, Chinese Academy of Sciences, Shenzhen, Guangdong, China. [5] National Synchrotron Radiation Laboratory, University of Science and Technology of China, Hefei, China. [6] International Center for Quantum Materials, Peking University, Beijing, China. [7] Electron Microscopy Laboratory, School of Physics, Peking University, Beijing, China. [8] Department of Engineering Mechanics, School of Aeronautics and Astronautics, Zhejiang University, Hangzhou, Zhejiang, China. [9] Applied Mechanics and Structure Safety Key Laboratory of Sichuan Province, School of Mechanics and Engineering, Southwest Jiaotong University, Chengdu, Sichuan, China. [10] Beijing National Laboratory for Condensed Matter Physics and Institute of Physics, Chinese Academy of Sciences, Beijing, China. [11] Collaborative Innovation Centre of Quantum Matter, Beijing, China. [12] State Key Laboratory for Artificial Microstructure and Mesoscopic Physics, School of Physics, Peking University, Beijing, China. [13] Shenzhen Key Laboratory of Quantum Science and Engineering, Shenzhen, Guangdong, China. [14] Key Laboratory of Soft Machines and Smart Devices of Zhejiang Province, Zhejiang University, Hangzhou, Zhejiang, China. [15] Interdisciplinary Institute of Light-Element Quantum Materials and Research Center for Light-Element Advanced Materials, Peking University, Beijing, China. [16] Guangdong Provincial Key Laboratory of Functional Oxide Materials and Devices, Southern University of Science and Technology, Shenzhen, Guangdong, China. [17] These authors contributed equally: Congbing Tan, Yongqi Dong, Yuanwei Sun, Chang Liu. ✉email: jw@zju.edu.cn; p-gao@pku.edu.cn; zlluo@ustc.edu.cn; lijy@sustech.edu.cn

Ferroelectric materials with spontaneous polarization have long been explored for information processing and data storage[1–3], and one of the defining characteristics of ferroelectrics is their polar domain configurations[4]. Unlike their ferromagnetic counterpart, ferroelectrics usually exhibit strong anisotropy that favors particular polar axes, and polarization rotation rarely occurs[5]. As a result, vast majority of polar structures naturally emerging in a ferroelectric are topologically trivial, and nontrivial polar topologies such as closure-domain[6,7], vortex[8,9], skyrmion[10], meron[11], and vortex-antivortex pair[12] have only recently been observed in reduced dimensions. It is now understood that in nanostructures such as nanodisks[13], nanorods[14], nanodots[15], nanoislands[16–20], and nanoplates[21], electrostatic force dominates polar crystalline anisotropy, forcing polarization rotation and thus the formation of vortex. For ultrathin superlattice subjected to misfit strain, such polar topologies can be alternatively driven by the delicate electromechanical competitions[22,23]. A recent study has observed the coexistence of flux-closure and $a_1/a_2$ strip-like domain in tensile-strained PbTiO$_3$(PTO) sandwiched between GdScO$_3$ substrate and a SrTiO$_3$(STO) layer, while their transformation under electron beam illumination has also been demonstrated[24]. These works raise the prospect of engineering topologically nontrivial polar topologies from topologically trivial domain architecture, which we seek to accomplish.

With the emergence of big data and ever increasing power consumption in microelectronics, nontrivial polar topologies become attractive as they promise alternative device configurations for high-density data storage[25] as well as ultralow power negative capacitance field effect transistors[26–29]. From an application point of view, it is highly desirable if we can engineer nontrivial polar topologies from the ordinary ferroelectrics with topologically trivial domain architecture, and with better understanding on the energetics of polar topologies[12,22], we are now in a position to explore such possibilities. Using a combination of techniques for polarization mapping[30], atomic imaging[6,7], and three-dimensional reciprocal space mapping (3D-RSM)[31] supported by phase-field simulations[32], we demonstrate the feasibility of engineering nontrivial polar topologies from topologically trivial domain architecture, and offer a realistic strategy to accomplish it. In particular, we have successfully reconstructed topologically trivial strip-like domain architecture into array of polar vortex in a model (PTO)$_{10}$/(STO)$_{10}$ (The subscript denotes 10-unit cell thickness) superlattices.

## Results

**Topologically trivial domain architecture of (PTO)$_{10}$/(STO)$_{10}$ superlattice.** Guided by the previous report on strip-like domain configuration[24,33,34], 17 cycles of (PTO)$_{10}$/(STO)$_{10}$ superlattice was synthesized on SrRuO$_3$ (SRO)-buffered DyScO$_3$ (001)$_{pc}$ (where pc refers to the pseudocubic notation) substrate using pulsed laser deposition (PLD), with details of the growth process provided in "Methods". X-ray diffraction (XRD) data in Supplementary Fig. 1a reveals the presence of *Pendellösung* fringes, confirming the high quality (PTO)$_{10}$/(STO)$_{10}$ superlattice with a smooth interface, while atomic force microscopy (AFM) topography mapping in Supplementary Fig. 1b shows the array of terraces with ~0.4 nm-height steps, further supporting high quality of the film. Lateral piezoresponse force microscopy (LPFM) was performed to map the ferroelectric domain structure as schematically shown in Supplementary Fig. 2, which measures the shear-induced torsional displacement of the cantilever arising from in-plane polarization perpendicular to the cantilever axis. Scanning under at least two orthogonal directions is thus

necessary to determine the in-plane polarization components, and we scanned by angles of 0°, 45°, and 90° between the cantilever axis and [010]$_{pc}$ direction. The resulted LPFM amplitude and phase mappings in Fig. 1a, b reveal a periodic strip-like domain configuration with domain wall oriented along the [110]$_{pc}$ direction, and no obvious contrast is seen in vertical PFM mapping over a larger area (Supplementary Fig. 3), indicating only in-plane polarization components. It is revealing to note that LPFM amplitudes scanned along 0° and 90° demonstrate opposite trend, with a peak in one direction corresponding to the valley in the other, which is evident in the line profile in Fig. 1c, while the period seen at 45° is only half of that seen at 0°- and 90°-scanning. Furthermore, 180° phase contrast is seen at 45°-scanning (Fig. 1d), while phase variation seen at 0°- and 90°-scanning is insignificant. These observations suggest alternating head-to-tail $a_1/a_2$ domain configuration as illustrated in Fig. 1e, which is topologically trivial. In particular, both $a_1$ and $a_2$ domains contribute to LPFM amplitude when scanned at 45°, reducing the apparent period in amplitude by half. Domain widths for both $a_1$ and $a_2$ domains can be estimated from the LPFM mapping over a larger area in Supplementary Fig. 3 along the line profile shown, as presented in Fig. 1f, from which the period of the strip-like domain is estimated to be 68.2 ± 2.6 nm. Such domain pattern is quite common in ferroelectrics, for example in strained Pb$_x$Sr$_{1-x}$TiO$_3$[35] and PbTiO$_3$ thin films[36], yet it is different from the vortex array seen in (PTO)$_{16}$/(STO)$_{16}$ superlattice[8], which we believe arises from different thicknesses. It will be exciting, however, if we can engineer such topologically trivial domain architecture into more interesting topologies, such as polar vortex.

**Emergence of polar vortex in (PTO)$_{10}$/(STO)$_{10}$ lamella.** In order to engineer possible polar topology of (PTO)$_{10}$/(STO)$_{10}$ superlattice, we fabricated thin lamellas from superlattice thin film as schematically shown in Supplementary Fig. 4. For the following transmission electron microscopy (TEM) studies, we focused on a region of lamella with thickness in the range of ~20–40 nm determined using the electron energy loss spectroscopy (EELS) method[37], as detailed in Supplementary Fig. 5. Low-magnification annular dark field (ADF) cross-sectional scanning transmission electron microscopy (STEM) image in Fig. 2a taken along the [010]$_{pc}$ zone axis reveals 17 stacking of PTO and STO layers with different contrast and approximately 8.0 nm period, and the thickness of the superlattice is measured to be 136 nm. The ADF-STEM imaging shows relatively sharp and coherent interfaces between PTO and STO, which is also confirmed by the atomically resolved energy dispersive spectrometry (EDS) mapping in Supplementary Fig. 6. Higher magnification ADF-STEM image in Supplementary Fig. 7 shows slight thickness variation in PTO layers at the unit-cell scale, though the typical configuration is determined as (PTO)$_{10}$/(STO)$_{10}$. We then acquired the dark field TEM (DF-TEM) image under two-beam conditions by selecting 002$_{pc}$ $g$-vector as shown in Fig. 2b, exhibiting periodic array of intensity modulation along the [100]$_{pc}$ direction with a spacing of ~8.6 nm. This DF-TEM image suggests a periodic distribution of clockwise-anticlockwise vortex pairs in PTO layers, as previously reported[8,38,39], which is also supported by the selected area electron diffraction (SAED) pattern in Fig. 2c acquired from a region that only includes the (PTO)$_{10}$/(STO)$_{10}$ superlattice layers. It reveals superlattice reflections both in-plane and out-of-plane, from which the in-plane lattice parameter $a$ and out-of-plane lattice parameter $c$ are calculated to be ~3.9 Å and ~4.0 Å, similar to those observed in reference 40[40]. In particular, the enlarged view of satellite diffraction spots around (001) reflection (inset in Fig. 2c) presents the long-range order in both

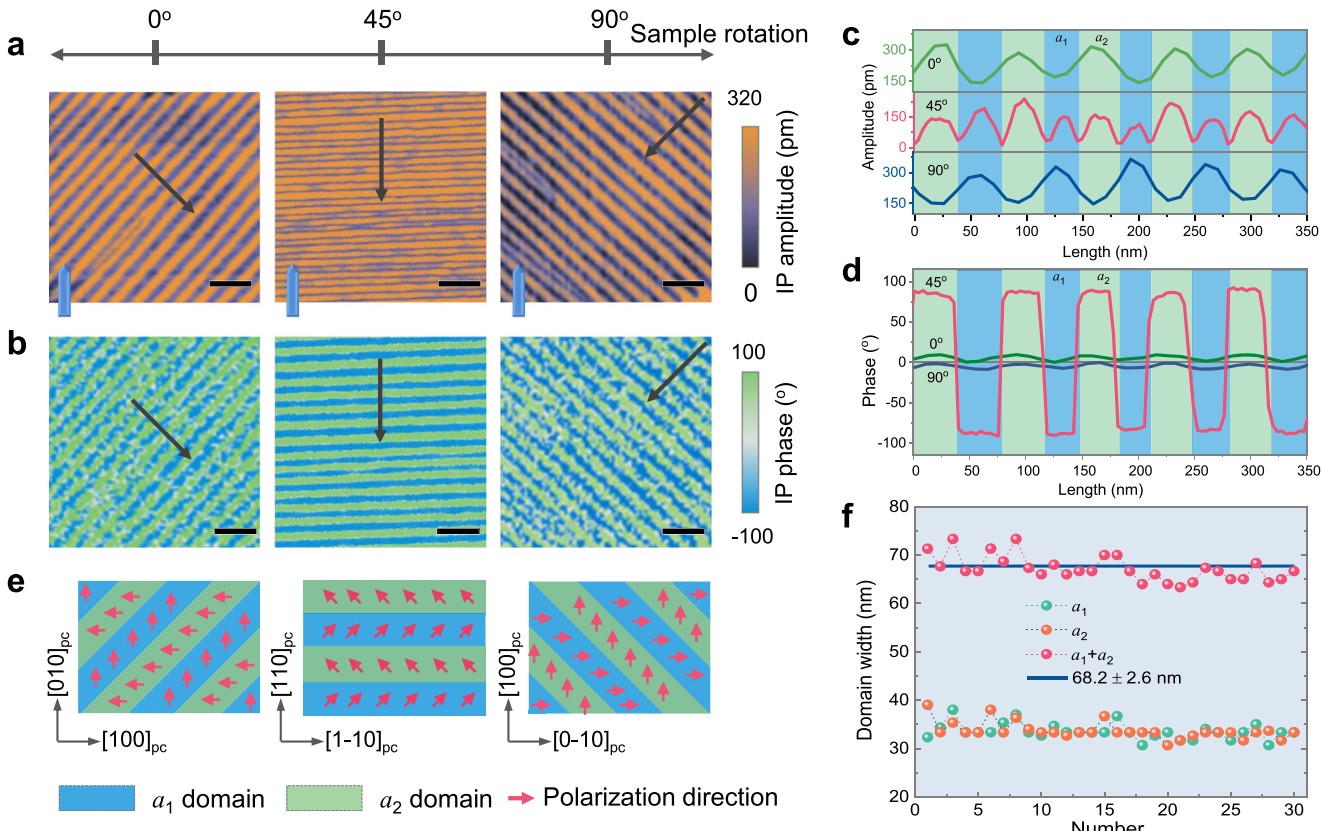

**Fig. 1 Periodic domain architecture of (PTO)$_{10}$/(STO)$_{10}$ superlattices. a**, **b** Lateral PFM amplitude and phase mappings of an $1 \times 1$ µm$^2$ areas measured at 0° (left), 45° (middle) and 90° between the cantilever axis and [010]$_{pc}$ direction with 200 nm scale bar. The blue obelisk-shaped marker denotes the orientation of PFM cantilever. **c**, **d** Amplitude and phase profiles along the black arrow line in (**a**, **b**). **e** Schematic of ferroelectric architecture with alternating $a_1$/$a_2$ domains separated by 90° domain walls. **f** Variation of domain widths along the line profile in Supplementary Fig. 3, with which the period of domain architecture is estimated to be 68.2 ± 2.6 nm.

out-of-plane and in-plane directions. The equal space in the out-of-plane direction is measured to be ~8.0 nm reflecting the period of the (PTO)$_{10}$/(STO)$_{10}$ superlattices. Another set of reflections along the in-plane direction indicates long-range ordering of vortices with a period of ~8.6 nm. In order to further confirm the existence of polar vortex, the detailed atomic polar displacement was mapped by calculating the offsets between A (Pb and Sr) and B (Ti) site sublattices based on the ADF-STEM image[6], and it can be clearly seen that polar vectors in the PTO layer rotate continuously around each vortex core (Fig. 2d), with alternating clockwise and anticlockwise orientations. We also calculated the corresponding tetragonality in terms of $c/a$ ratio (Fig. 2e), wherein the red sinusoidal wave-like pattern also suggests the vortex pair as previously reported[8,38–40]. Pairs of clockwise and anticlockwise vortices are schematically shown in Fig. 2f, with their position taken from Fig. 2e, and the distance between the two adjacent anticlockwise vortices is estimated to be ~8.6 nm according to DF-TEM and SAED data. To examine the effect of substrate anisotropy, thin lamella was also cut from superlattice film in a perpendicular direction, and the resulted cross-sectional DF-TEM (supplementary Fig. 8) taken along the orthogonal [100]$_{pc}$ zone axis also shows periodic contrast modulation along the [010]$_{pc}$ direction. The corresponding electron diffraction pattern contains similar satellite spots as well, indicating the presence of periodic vortex arrays independent of lamella orientation with respect to the substrate. It is thus established that arrays of the polar vortex have been engineered in (PTO)$_{10}$/(STO)$_{10}$ lamellas from topologically trivial domain architecture in (PTO)$_{10}$/(STO)$_{10}$ superlattice film.

**3D domain architecture of (PTO)$_{10}$/(STO)$_{10}$ superlattice.** Piezoresponse force microscopy can only reveal polar structure near the sample surface, while 3D domain architecture can be distinct from the surface configuration. And thus there exists a possibility that polar vortex in (PTO)$_{10}$/(STO)$_{10}$ superlattice not recognized by PFM. To resolve this, we resort to synchrotron-based XRD as illustrated in Fig. 3a. The diffraction intensity around 002$_{pc}$ Bragg spot for the 17 cycles of (PTO)$_{10}$/(STO)$_{10}$ superlattice deposited on SRO-buffered DyScO$_3$ was obtained via 3D-RSMs as shown in Fig. 3b, and there are no in-plane periodic satellites observed beside the PTO peak, suggesting the absence of polar vortex. As exemplified in Supplementary Fig. 9, in-plane satellite peaks always accompany the Bragg spot of the PTO reflections in (PTO)$_{20}$/(STO)$_{20}$ superlattice possessing periodic vortex array[31]. A Q$_{[h00]}$-Q$_{[00l]}$ slice from this 002$_{pc}$ 3D-RSM, as shown in Fig. 3c, reveals a single alloy film peak and superlattice reflections (marked as SL (±n)). The location of alloy peak quantitatively confirms alternating $a_1$/$a_2$ domains in PTO with an out-of-plane lattice constant of 3.915 Å, consistent with the previous reports on $a_1$/$a_2$ domain structure[31,40]. The 15 fringes observed between superlattice peaks in Fig. 3d indicate a total of 17 cycles of (PTO)$_{10}$/(STO)$_{10}$, and the out-of-plane superlattice interval periodicity is measured to be 7.9 nm. Furthermore, 3D-RSM around 103$_{pc}$ of the DyScO$_3$ substrate and the corresponding planar slice containing [−kk0]$_{pc}$ and [00l]$_{pc}$ are shown in Fig. 3e, f, wherein a set of significant satellites along the in-plane <110>$_{pc}$ are observed, corresponding to a ~70 nm periodicity of $a_1$/$a_2$ domains along this direction in PTO, in good agreement with PFM data in Fig. 1f. Note that the structural modulation in $a_1$/$a_2$

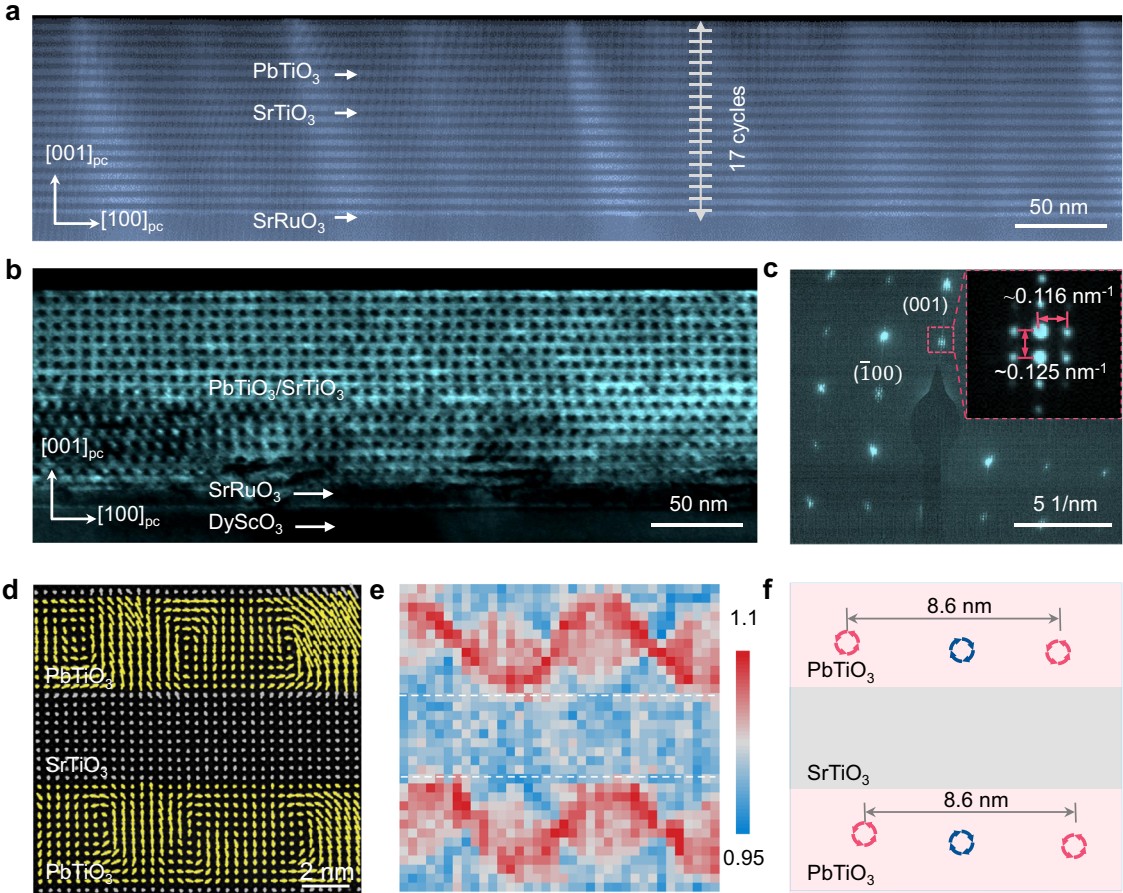

**Fig. 2 Polar vortex arrays in (PTO)$_{10}$/(STO)$_{10}$ superlattices. a** Cross-sectional low-magnification ADF-STEM images, with 50 nm scale bar. **b** DF-TEM image under two-beam conditions by selecting 002$_{pc}$ g-vector, with 50 nm scale. The periodic array of bright and dark intensity modulation corresponds to vortex arrays within PTO layers. **c** SAED pattern, with 5 nm$^{-1}$ scale bar. Inset: enlarged (001) reflection showing the satellite diffraction spots. The intervals of diffraction points in out-of-plane and in-plane directions are measured to be ~0.125 and ~0.116 nm$^{-1}$, respectively, indicating the periods of superlattice and vortex arrays to be ~8.0 and ~8.6 nm. **d** Polar vector map illustrating the clockwise and anticlockwise vortices in PTO layers, with 2 nm scale bar. **e** The corresponding tetragonality (c/a ratio) mapping. The dashed lines denote the interfaces. **f** The schematic illustration of clockwise (blue circles) and anticlockwise (red circles) vortices in the same position as shown in (**d**).

domains is completely in-plane, therefore it does not produce satellites beside any Bragg spot along the vertical [00l]$_{pc}$ direction. We also acquired the planar-view DF-TEM image of the (PTO)$_{10}$/(STO)$_{10}$ superlattice film as shown in Supplementary Fig. 10, which exhibits long-range strip-like domain along the [110]$_{pc}$ direction with the period estimated to be ~69 nm consistent with both PFM and 3D-RSM data. There exist a few strip-like domains with narrower periodicity, and there are a small number of strip-like domains with perpendicular orientation along [−110] as well (Supplementary Fig. 10d), which can also be observed by PFM over a larger area (Supplementary Fig. 10e). These topologically trivial domain patterns are fully consistent with continuum analysis based on strain and polarization compatibilities[4], and it is distinct from the orientations and period of vortex array reported in ref. [8]. These set of data thus reconfirms the topologically trivial strip-like domain structure in (PTO)$_{10}$/(STO)$_{10}$ superlattice before reconstruction.

**Phase-field simulation of polar vortex reconstruction.** It is evident from 3D-RSM, PFM and STEM data that polar reconstruction occurs when the thin lamella is fabricated, and in order to understand the process, phase-field simulations were carried out as shown in Fig. 4. Under a biaxial strain of

$\varepsilon_{[100]} = \varepsilon_{[010]} = -0.3\%$, the superlattice forms topologically trivial strip-like domains in PTO layers consistent with PFM and 3D-RSM data presented in Figs. 1, 3. Alternating $a_1/a_2$ domains separated by 90° domain walls along the [110]$_{pc}$ direction are found (Fig. 4a–c), while the out-of-plane polarization is negligible. When a lamella of superlattice is cut from thin film along the [010]$_{pc}$ direction (Fig. 4d), the biaxial strain is converted to a uniaxial one with $\varepsilon_{[010]}$ released. Then nontrivial polar structure with alternating clockwise and anticlockwise vortex distribution along the [100]$_{pc}$ direction emerges in PTO (Fig. 4d–f), with $P_{[001]}$ increasing at the expense of $P_{[010]}$. It is also insightful to compare the out-of-plane lattice constant of PTO in superlattice film (3.915 Å via 3D-RSM) and lamella (4.0 Å via SAED), supporting the reconstruction of polar configuration. With out-of-plane polarization suppressed in strip-like $a_1/a_2$ domains, the corresponding lattice constant is inevitably compressed. This demonstrates that polar vortex is rather sensitive to electromechanical conditions. Indeed, by applying a larger biaxial strain of $\varepsilon_{[100]} = \varepsilon_{[010]} = -1\%$ to the superlattice, polar vortex also emerges in superlattice film (Fig. 4g–i) without going into uniaxial lamella configuration. Here the vortex array extends along the [110]$_{pc}$ direction instead, and a detailed phase diagram of polar topologies is presented in Supplementary Fig. 11.

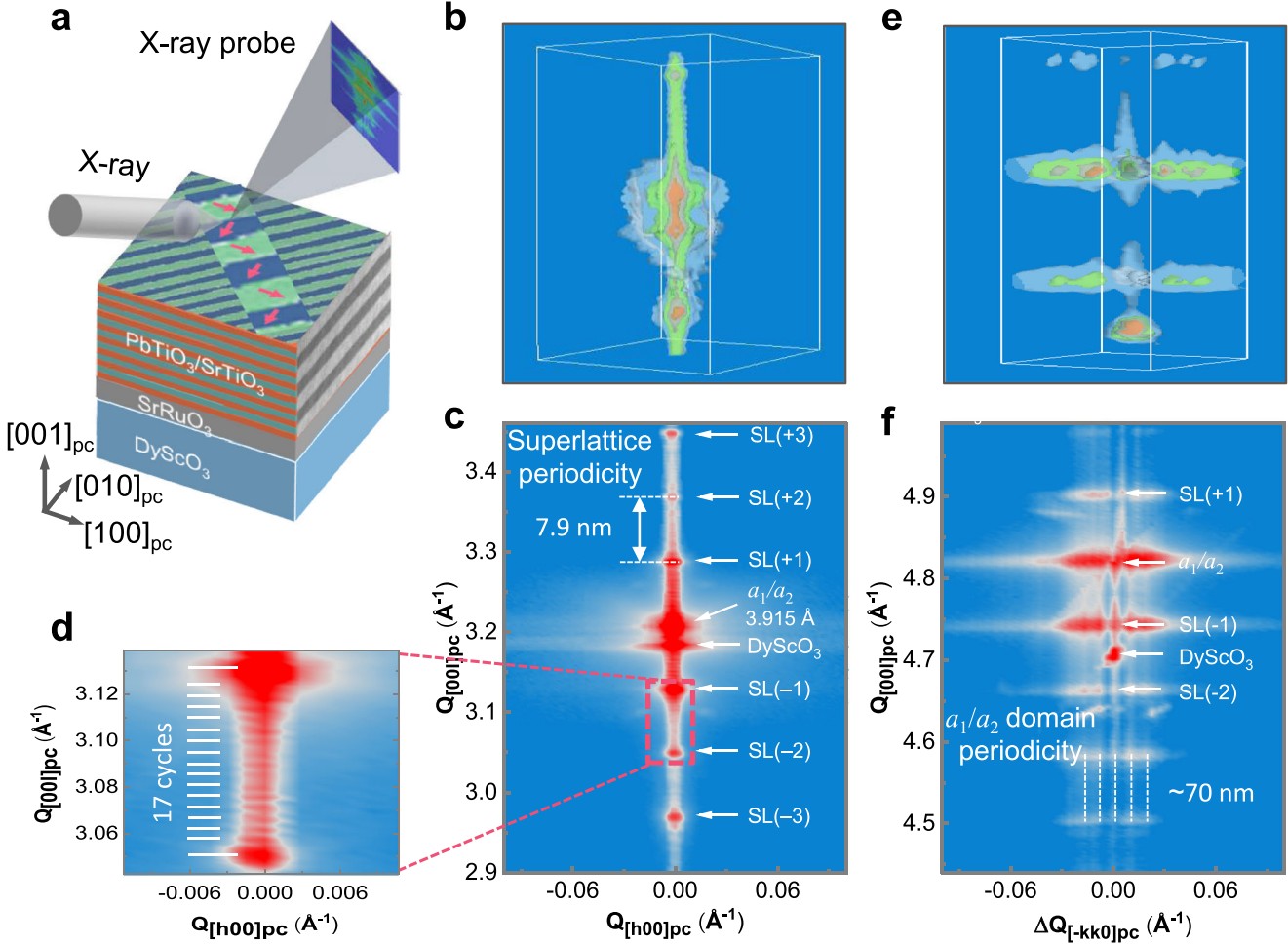

**Fig. 3 3D-RSM characterization of the (PTO)$_{10}$/(STO)$_{10}$ superlattice. a** Schematic of X-ray diffraction of (PTO)$_{10}$/(STO)$_{10}$ superlattice. **b** 3D-RSM around 002$_{pc}$ Bragg spot of the DyScO$_3$ substrate and (**c, d**) the corresponding planar slice along Q$_{[h00]}$-Q$_{[00l]}$, showing the reflections of PTO $a_1$/$a_2$ domains and stacking superlattice. **e** 3D-RSM around 103$_{pc}$ Bragg spot of the DyScO$_3$ substrate and (**f**) the corresponding planar slice along Q$_{[-kk0]}$-Q$_{[00l]}$ showing the in-plane diffractive satellites along <−kk0> direction.

## Discussion

We can understand the reconstruction of topologically trivial domain configuration into polar vortex from energetic point of view, resulted from the competition among the electrostatic energy, the elastic energy and the gradient energy[41,42]. The electrostatic energy is due to the depolarization field, and when the lamella is cut from thin film, large discontinuity in polarization normal to the lamella surface is generated, preventing the out-of-lamella polarization in PTO. The remaining uniaxial strain prefers the original out-of-plane polarization, while electrostatic energy discourages it, and the competition results in the formation of vortex with delicate balance, which also increases the out-of-plane lattice constant as observed. The key lies in the reduced dimension, otherwise ordinary domain patterns can be easily formed to accommodate different requirements[4]. Such understanding can guide us exploring as well as engineering polar topologies further. For example, we can explore planar instead of cross-sectional lamella that may modify the magnitude of biaxial misfit strain and thus change the energy landscape, which is currently under investigation.

With respect to potential applications, the polar vortex has been suggested for high-density data storage, yet in a thin film configuration, the vortex structure is usually one-dimensional in nature, as exhibited by Fig. 4g–i, which reduces the storage density substantially. Our lamella configuration overcomes such difficulty by accommodating arrays of zero-dimensional polar vortex, as exhibited by Fig. 4d–f. Other applications such as negative capacitance field effect transistors can also be envisioned. On more fundamental side, we demonstrate that such polar vortex can be engineered from topologically trivial domain configuration, expanding the reach of topological polar structures into the otherwise ordinary ferroelectric superlattice. In this regard, our work also highlights the importance of combining surface probing, advanced microscopies, and 3D structural characterization to accurately determine the polar topologies. Reconstruction can occur, and thus information from a single microscopy technique could be incomplete.

## Methods

**Fabrication of (PTO)$_{10}$/(STO)$_{10}$ superlattices.** Superlattices of (PTO)$_{10}$/(STO)$_{10}$ (where 10 denote the thickness of alternating stacks of the STO and PTO layers in term of unit cells) were fabricated on 12 nm SRO-buffered (110)-DyScO$_3$ substrates in a pulsed laser deposition system (PVD-5000) equipped with a KrF excimer laser (λ = 248 nm). The SRO-buffered layer was first grown at a 690 °C substrate temperature and 80 mTorr oxygen pressure. Then, the substrate temperature was cooled to 600 °C and the oxygen pressure went up to 200 mTorr for the subsequent alternating growth of the PTO and STO layers. The laser energy for the growth of the SRO and the (PTO)$_{10}$/(STO)$_{10}$ superlattices was 400 and 360 mJ pulse$^{-1}$, respectively. Selecting the correct laser energy was crucial for fully strained PTO and STO sublayers, which is a prerequisite for the presence of polar vortices in the superlattice. Thicknesses of the SRO, PTO, and STO layers were held to be 30-, 10-, and 10-unit cells, respectively, by controlling the growth time. After growth, the

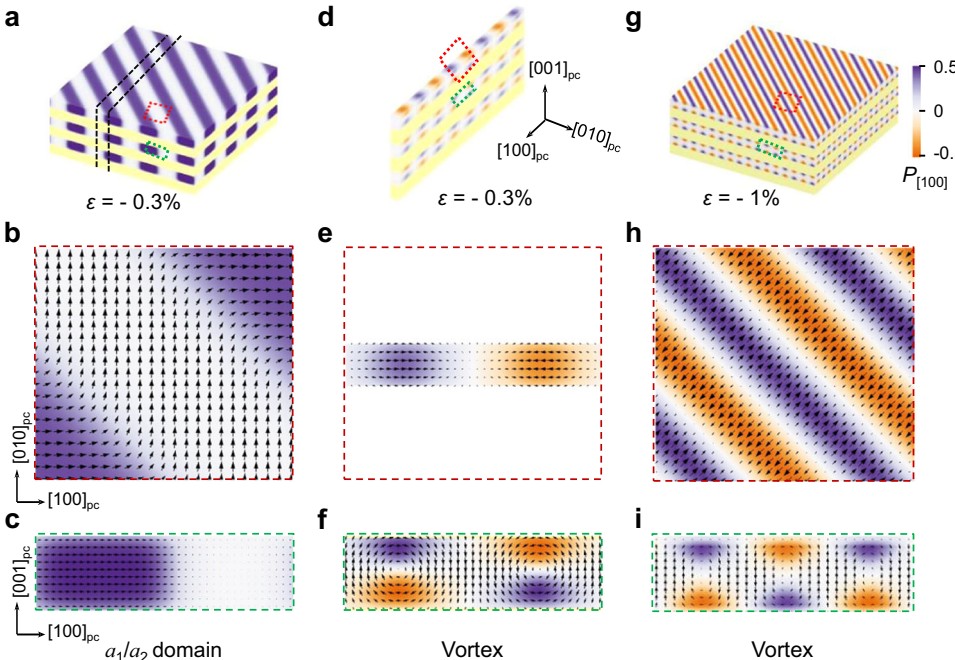

**Fig. 4 Phase-field simulation of (PTO)₁₀/(STO)₁₀ superlattice. a** Spatial distribution of polarization with equiaxial strain $\varepsilon = -0.3\%$, showing the $a_1/a_2$ domain structure. Periodic boundary conditions are applied along $[100]_{pc}$, $[010]_{pc}$ and $[001]_{pc}$ directions, respectively. **b, c** Zoom-in results corresponding to the area labeled by red and green rectangles in (**a**). **d** The slices cut from (**a**) within the black dotted lines, showing reconstructed vortex structure. The thickness of the slices is 2.4 nm. **e, f** Zoom-in results for the vortex structure labeled by the red and green rectangles in (**d**), respectively. **g** Spatial distribution of polarization with biaxial strain $\varepsilon = -1\%$, showing the vortex structure. **h, i** Zoom-in results corresponding to the representative areas labeled by red and green rectangles, respectively.

(PTO)₁₀/(STO)₁₀ superlattice films were slowly cooled to room temperature at a 200-mTorr oxygen pressure.

**Atomic force and piezoresponse force microscopies**. The (PTO)₁₀/(STO)₁₀ superlattices were characterized using a commercial scanning probe microscope (Asylum Research, MFP-3D infinity) in tapping mode to access the morphology, and in the dual AC resonance tracking (DART) piezoresponse mode (both in-plane and out-of-plane modes) to record the ferroelectric domain structure. Pt conductive tips with a force constant of 10 N m⁻¹ (RMN-25PT400B) were used for piezoresponse force microscopy measurement. Since in-plane (IP) PFM only measures the component perpendicular to the cantilever axis, which is associated with the torsional vibration of the cantilever, we gathered the real part images of IP PFM signals measured at different probe orientations to construct a full vector map of the IP piezoresponse vector.

**Lamella preparation**. The planar-view and cross-sectional lamella specimens were thinned to ~30 μm by using mechanical polishing or cut via focused ion beam (FIB), and then by performing argon ion milling. The argon ion milling was carried out using the Precision Ion Polishing System with accelerating voltage of 3.5 kV and then stepped down to a final cleaning voltage of 0.3 kV.

**Dark field and ADF images**. Diffraction contrast TEM experiments were carried out using JEOL ARM 300 F instrument at 300 kV. Dark field imaging was performed in the two-beam condition with the samples being tilted off the zone axis and imaged in the dark field using 002_pc **g** vector. ADF-STEM images were recorded from an aberration-corrected FEI Titan Themis G2 at 300 kV. The convergence semi-angle for imaging was 30 mrad, and the collection semi-angle was 39–200 mrad. The atom positions were determined by simultaneously fitting two-dimensional Gaussian peaks to an a priori perovskite unit cell using a MATLAB code. The polar vectors were plotted from the offset between A (Pb and Sr) and B-site (Ti) sublattices based on the ADF-STEM image in Fig. 2d. We acquired the ADF-STEM images with a smaller convergence semi-angle for enhancing the stress contrast in the strained PTO layer. Different convergence semi-angle will only affect the relative intensity of the atomic column, not the position of the atom columns.

**Thickness measurement of the lamella**. The lamella thickness was estimated using the STEM-EELS method. STEM-EELS was performed using an aberration-corrected FEI Titan Themis G2 at 300 kV. The typical energy resolution (half-width of the full zero-loss peak, ZLP) was 0.8 eV. For the different regions of

superlattice lamella, STEM-EELS images in the low loss region were acquired and converted to a thickness map of inelastic mean free paths ($t/\lambda$) in Gatan's Digital Micrograph software using the absolute log-ratio method for the zero-loss and the first plasmon peaks[37].

**Synchrotron-based X-ray diffraction**. The 3D-RSM obtain from synchrotron X-ray diffraction were employed to study the complex phase coexistence in the superlattice structures. The 3D-RSM experiments were performed at the 33-BM-C beamline of the Advanced Photon Source and BL14B beamline of Shanghai Synchrotron Radiation Facility. High accurate 3D-RSMs were obtained by the excellent accuracy of Huber 6-circle diffractometer equipped with Pilatus 100K or Eiger X 500K area detector.

**Phase-field simulation**. In the phase-field model, the free-energy density $f$ of the STO/PTO superlattice has the following form

$$f = \alpha_i P_i^2 + \alpha_{ij} P_i^2 P_j^2 + \alpha_{ijk} P_i^2 P_j^2 P_k^2 + \frac{1}{2} c_{ijkl} \varepsilon_{ij} \varepsilon_{kl} - q_{ijkl} \varepsilon_{ij} P_k P_l$$
$$+ \frac{1}{2} g_{ijkl} P_{i,j} P_{k,l} - \frac{1}{2} \varepsilon_0 \varepsilon_r E_i E_i - E_i P_i, \tag{1}$$

in which $\alpha_i$, $\alpha_{ij}$ and $\alpha_{ijk}$ denote the Landau expansion coefficients, $c_{ijkl}$ are the elastic constants, $q_{ijkl}$ are the electrostrictive coefficients, $g_{ijkl}$ are the gradient coefficients, $\varepsilon_0$ is the vacuum permittivity and $\varepsilon_r$ represents the relative dielectric coefficient. $P_i$, $\varepsilon_{ij}$ and $E_i$ are the polarization, strain and electric field components, respectively. All repeating subscripts in Eq. (1) imply summation over the Cartesian coordinate components $x_i$ ($i = 1, 2$ and $3$), and subscript ",$i$" denotes the partial derivative operator with respect to $x_i$ ($\partial/\partial x_i$).

The temporal evolution of the polarization and domain structure can be described by the time-dependent Ginzburg–Landau (TDGL) equation as

$$\frac{\partial P_i(\mathbf{r}, t)}{\partial t} = -L \frac{\delta F}{\delta P_i(\mathbf{r}, t)} \tag{2}$$

where $L$ represents the kinetic coefficient that controls the speed of domain evolution; $F = \int_V f \, dV$ is the total free energy in the whole structure; $\mathbf{r}$ and $t$ denote the spatial position vector and time, respectively. In addition, both the mechanical equilibrium equation

$$\sigma_{ij,j} = \frac{\partial}{\partial x_j}\left(\frac{\partial f}{\partial \varepsilon_{ij}}\right) = 0, \tag{3}$$

**Table 1 The material constants of PTO and STO[43].**

| Landau coefficient | | Elastic coefficient | | Electrostriction coefficient | |
|---|---|---|---|---|---|
| $\alpha_1^{PTO}$ ($10^8 m^2 N\ C^{-2}$) | −1.7 | $c_{11}^{PTO}$ (GPa) | 230 | $Q_{11}^{PTO}$ ($m^4\ C^{-2}$) | 0.089 |
| $\alpha_{11}^{PTO}$ ($10^8 m^6 N\ C^{-4}$) | −0.73 | $c_{12}^{PTO}$ (GPa) | 100 | $Q_{12}^{PTO}$ ($m^4\ C^{-2}$) | −0.026 |
| $\alpha_{12}^{PTO}$ ($10^8 m^6 N\ C^{-4}$) | 7.5 | $c_{44}^{PTO}$ (GPa) | 70 | $Q_{44}^{PTO}$ ($m^4\ C^{-2}$) | 0.0675 |
| $\alpha_{111}^{PTO}$ ($10^8 m^{10} N\ C^{-6}$) | 2.6 | $c_{11}^{STO}$ (GPa) | 330 | $Q_{11}^{STO}$ ($m^4\ C^{-2}$) | 0.0457 |
| $\alpha_{112}^{PTO}$ ($10^8 m^{10} N\ C^{-6}$) | 6.1 | $c_{12}^{STO}$ (GPa) | 100 | $Q_{12}^{STO}$ ($m^4\ C^{-2}$) | −0.0135 |
| $\alpha_{123}^{PTO}$ ($10^8 m^{10} N\ C^{-6}$) | −37 | $c_{44}^{STO}$ (GPa) | 125 | $Q_{44}^{STO}$ ($m^4\ C^{-2}$) | 0.00957 |
| $\alpha_1^{STO}$ ($10^5 m^2 N\ C^{-2}$) | 2.017 | | | | |
| $\alpha_{11}^{STO}$ ($10^8 m^6 N\ C^{-4}$) | 1.7 | | | | |
| $\alpha_{12}^{STO}$ ($10^8 m^6 N\ C^{-4}$) | 4.45 | | | | |

For pure PTO and STO material at room temperature (25 °C), the free energy coefficients are given in Table 1. Additional data about the Landau coefficients at different temperatures can be found in Chen's report[44].

and Maxwell's equation

$$D_{i,i} = -\frac{\partial}{\partial x_i}\left(\frac{\partial f}{\partial E_i}\right) = 0, \qquad (4)$$

are satisfied simultaneously for a body-force-free and charge-free ferroelectric system, where $\sigma_{ij}$ and the $D_i$ are the stress and electric displacement component, respectively. The semi-implicit Fourier-spectral method[32] is employed to solve the partial differential equation. Three-dimensional simulation is carried out for clearer illustration and computational simplicity, where $3 \times 3$ periodic PTO/STO superlattices model is adopted here. And $150\Delta x \times 150\Delta x \times 60\Delta x$ grid point with $\Delta x = 0.4$ nm in real space is used for space discretization, where 10-grid is used along the thickness direction of each PTO and STO layer. And the length of time step is chosen as $\Delta t/t_0 = 0.2$, where $t_0 = 1/(\alpha_0 L)$ with $\alpha_0$ representing the absolute value of $\alpha_1$ at room temperature. Periodic boundary conditions for electric potential and polarization components are employed along the $[100]_{pc}$, $[010]_{pc}$, and $[001]_{pc}$ direction, respectively. The material parameters for PTO and STO used in the simulation are listed in Table 1. Small random fluctuation ($<0.01P_0$, where $P_0 = 0.76$ C m$^{-2}$ is the spontaneous polarization of PTO at room temperature) is used as the initial setup for polarization to initiate the simulation.

## Data availability
All data needed to evaluate the conclusions in the paper are present in the paper and/or the Supplementary Information. Additional data related to this paper may be requested from the authors.

## Code availability
The MATLAB-based toolbox for fitting atom positions and calculating local polarization is available from the corresponding author upon request.

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

## Acknowledgements

This work was supported by the National Key R&D Program of China (Grant Nos. 2016YFA0201001, 2016YFA0300102), the National Natural Science Foundation of China (Grant Nos. 51872251, 11875229, 11972320, 11675179, 11627801, and 92066203), the National Equipment Program of China (ZDYZ2015-1), the Key R&D Program of Guangdong Province (Grant Nos. 2018B030327001, 2018B010109009, and 2019B010931001), Shenzhen Science and Technology Innovation Committee (Grant Nos. JCYJ20170818163902553, JCYJ20200109115219157), the Leading Talents Program of Guangdong Province (2016LJ06C372), Guangdong Provincial Key Laboratory Program (2021B1212040001) from the Department of Science and Technology of Guangdong Province, Zhejiang Provincial Natural Science Foundation (Grant No. LZ17A020001), and the "2011 Program" Peking-Tsinghua-IOP Collaborative Innovation Center for Quantum Matter. We also acknowledge the Electron Microscopy Laboratory in Peking University for the use of the $C_s$-corrected electron microscope. Z.L. acknowledges the support from the beamline BL14B of Shanghai Synchrotron Radiation Facility (SSRF) and the beamline 33BM of Advanced Photon Source.

## Author contributions

J.L., C.T., P.G., J.W. and Z.L. conceived the idea and designed the work. C.T. grew the samples assisted by J.B.W., X.Z., M.L. and X.O; Y.S., C.P. and R.Z performed the electron microscopy experiments and assisted by M.W., K.L., X.B., D.Y. and J.Z. under the direction of P.G.; Y.D. and Z.L. carried out synchrotron structural characterization; C.L. did the phase-field simulation under the direction of J.W. and J.L.; C.T. and Y.S. performed data analysis assisted by Y.D. and C.L. under the direction of J.L., J.W., Z.L. and P.G.; C.T. and J.L. wrote the manuscript with the assistance of Y.W.S., Y.D. and C.L. and all the authors. All authors discussed the results and commented on the manuscript.

## Competing interests

The authors declare no competing interests.
