## [Peer Review File · Nature Communications]

REVIEWER COMMENTS

Reviewer #1 (Remarks to the Author):

Ref.: NCOMMS-21-11782-T by Tan et al.

Review report:

In this work, the authors present an interesting study on the manipulation of polar vortex phase in (PbTiO₃)₁₀/(SrTiO₃)₁₀ superlattice grown on DSO substrate, from a topological trivial a1a2 domains. The sample was grown with a very high quality, as seen in the atomic level EDS maps. This work is illuminating for future integrating of the wafer scale topological trivial domains into 3D nanostructures containing vortex phase, which would be useful for future employment of the polar topologies in oxides.

I think this work should be published. There are, however, a few concerns and I'd like to see them be addressed before final consideration of publication.

1. In this study, the superlattice that the authors claimed is (PbTiO₃)₁₀/(SrTiO₃)₁₀, however, according to the EDS maps shown in Supplementary Fig. 5, the superlattice is likely (PbTiO₃)₁₀-12/(SrTiO₃)₉ rather than (PbTiO₃)₁₀/(SrTiO₃)₁₀. Since the phase components of PTO/STO superlattice samples are very sensitive to layer thicknesses, the authors should accurately describe the geometry of the superlattice.
2. It is known that DSO substrate features anisotropy and usually it has significant effects on the domain structures of ferroelectric films grown on it, I suggest the authors make some discussions on how the vortex phase looks like if the sample is cut from the perpendicular direction.
3. In Page 6, the authors describe that ".....the in-plane lattice parameter a and out-of-plane lattice parameter c are calculated to be ~3.93 Å and ~4.02 Å". I suggest that such a statement should be rearranged, since electron diffraction cannot provide such a high precision for determination of lattice parameters.
4. Some of the description in the text should be tuned down, such as "From application point of view, it is highly desirable if we can engineer nontrivial polar topologies from the ordinary ferroelectrics with topological trivial domain architecture, yet this has not been accomplished to our best knowledge". Actually, observation of a1/a2 to flux-closure domain transformations induced by e-beam was reported recently (Acta Materialia 193 (2020) 311).
5. Finally, I have to point out that the references cited in this manuscript particularly in the abstract and Introduction sections are randomly showed up in order, and consequently the contextualization of this work is not good enough. When taking about topological ferroelectric domains, one cannot start from the Nature (2016) work (Reference No.1 in the present manuscript) while ignoring the study that was published one year before (Y.L. Tang et al. Science 2015). One can see that the sample system in the Nature (2016) paper is the same as that in the Science paper (2015), and the domain configurations are similar in these two papers. Overlooking experimental observation on topological ferroelectric domains, relevant studies should be discussed in a chronological order, such as Science (2015) on flux-closure, Nature (2016) on vortices, Nature (2019) on skyrmion, and Nature Materials (2020) on meron. I find, in a very recent publication by the present collaborative team (Nature Communications 2020), they presented a very nice review on the recent progress of topological domains. I suggest the authors should maintain a single style on reviewing the same topic. In the high-profile journals like nature communications, a well-presented contextualization on the previous studies in the field is necessary -- a good review on literature is a kind of contribution as well to the academic community.

Review of "Reconstruction of polar vortex from topological trivial domain architecture" by Congbing Tan et al. submitted to *Nature Communications*.
Manuscript ID: NCOMMS-21-11782-T

The manuscript reports that the topologically non-trivial polar structure (polar vortices in oxide superlattices $(\text{PbTiO}_3)_{10}/(\text{SrTiO}_3)_{10}$ as reported by *Yadav, A. K. et al. (Observation of polar vortices in oxide superlattices. Nature 530, 198-201 250 (2016))* are actually created by the reconstruction from a topologically trivial strip domain structure in the as-deposited film on a substrate. The reconstruction is induced by the thin plate fabrication for TEM observations.

As my first impression, the present manuscript could have been submitted to *Nature* as a **Comment on "Observation of polar vortices in oxide superlattices." by Yadav, A. K. et al. Nature 530, 198-201 (2016)** rather than an independent publication. It appears, however, that the authors chose the latter option. They wrote a manuscript claiming that the reconstruction is useful as an artificial engineering technique of topologically non-trivial polar vortices from topologically trivial stripe domain, instead of criticizing the previous publication. I would like to leave the editor to choose the best way to publish their findings.

Overall, their manuscript is well written based on technically sound experimental data with scientific rigor. Of course, the contents would attract a majority of professional and general readers as was proved by the previous *Nature* publication. So I recommend the manuscript to be published on a high-impact journal. However, their manuscript must be brushed up further by supplementing additional data to complement their analysis on this fascinating ferroelectric structure at the nanometer length-scale.

As a conclusion, I would like to reconsider the manuscript for a publication in *Nature Communications* after major and minor revisions as I describe in the following paragraphs.

Major revisions

(1) A plan-view DF-TEM image as shown in Fig. 3b in Ref. 1 must be presented. The data is significant to discuss how the two orthogonal thin plate fabrication processes reconstruct the original stripe domain.

(2) PFM measurements reveal the period of stripe domain is 68.2 ± 2.6 nm, which is much different from the period of vortices (8.6 nm) as well as their phase-field simulations (also 8.6 nm?). The authors must explain the discrepancy in the main text. In addition, the orientation of the stripe domain (a-c domain) is different from the stripe (a-a domain) as shown in Fig. 3b and Fig. 4 in the Ref. 1. The authors must explain the difference in the main text as well.

(3) Introduction is too short (only 225 words). Please refer to *Abid, A. Y. et al. Creating polar antivortex in $\text{PbTiO}_3/\text{SrTiO}_3$ superlattice. Nature Communications 12, 2054, doi:10.1038/s41467-021-22356-0 (2021)*. A typical Introduction is composed of three paragraphs. It would be a good idea to include this latest publication in Introduction as it appears closely related with the present work.

(4) Also, Discussion is too short (only 122 words). It would be a good idea to add a discussion on the difference between the plan-view lamella and the cross-sectional

lamella. The readers must be interested in how the two engineering processes reconstruct the original stripe domain.

Minor Revisions

- (0) Throughout the manuscript: "topological trivial" -> "topologically trivial"
- (1) Title: "Reconstruction of polar vortex from topological trivial domain architecture" -> What is reconstructed is not the polar vortex but the topologically trivial domain. "Reconstruction (or Engineering) of topologically trivial stripe domain into topologically non-trivial polar vortex" would be better.
- (2) Abstract, line 31: "Topological polar structures" -> "Topologically non-trivial polar structures"
- (3) Abstract, line 35: "artificially engineering nanostructured polar topologies" -> "artificially engineered polar topologies" as vortex, skyrmion, and meron possess non-trivial topology.
- (4) Abstract, line 36: "reconstruction of topological trivial strip-like domain architecture commonly observed into arrays of polar vortex" -> "reconstruction of topologically trivial strip-like domain architecture into arrays of polar vortex"
- (5) Line 40: "polarization mapping⁷, atomic imaging⁸, and three-dimensional (3D) structure visualization⁹" -> "polarization mapping by atomic imaging^{7,8} and three-dimensional (3D) reciprocal space mapping⁹"
- (6) Line 43: "tipping" -> The meaning is vague. Use a more comprehensive phrase.
- (7) Between line 46 and 47: Add "Introduction".
- (8) Line 58: "From application point of view" -> "From an application point of view"
- (9) Line 76: Insert "Results".
- (10) Line 76: "Guided by previous report on strip-like domain configuration" -> A citation to the "previous report" is necessary.
- (11) Figure 2, caption, line 112: "scanning transmission electron microscopy (STEM) image" -> "ADF-STEM image or HAADF-STEM image", All acronyms must be defined in the main text when they appear first.
- (12) Figure 2, caption, line 113: "Dark field TEM image" -> "DF-TEM image", All acronyms must be defined once in the main text when they appear first.
- (13) Figure 2, caption, line 115: "A selected-area electron diffraction (SAED) pattern" -> "SAED pattern", All acronyms must be defined once in the main text when they appear first.
- (14) Line 136: "dark field TEM" -> "dark field TEM (DF-TEM)"
- (15) Line 155: "Fig. 3f" -> "Fig. 2f", "Fig. 3e" -> "Fig. 2e" (?)
- (16) Line 157: "dark field TEM" -> "DF-TEM"
- (17) Line 239: "Our lamella configuration overcome such difficulty by accommodating arrays of zero-dimensional polar vortex, as exhibited by Fig. 4d-f." -> Add more explanation on the usefulness of their engineering technique if possible.

End of Review

Reviewer #3 (Remarks to the Author):

This manuscript shows that a $(\text{PbTiO}_3)_{10}/(\text{SrTiO}_3)_{10}$ superlattice sample comprises “topologically trivial” strip-like polarisation domains as fabricated but that a thin lamella prepared from this sample comprises an array of polarization vortices. As regards the electron microscopy, the analysis uses standard techniques for such systems but is thorough in showing that multiple forms of evidence point to a common conclusion. I leave any critique warranted on the details of the piezo-response force microscopy, X-ray 3D reciprocal space mapping and phase field simulations to reviewers better qualified to make it, observing only that I found the description of these analyses to be clear.

The point of novelty of the manuscript seems to me somewhat subtle. It is not that $(\text{PbTiO}_3)_{10}/(\text{SrTiO}_3)_{10}$ superlattice thin films may contain an array of polarization vortices: as the authors acknowledge, these have been observed via atomic-resolution electron microscopy before (e.g. Ref. [1], and by some of the present authors in Ref. [29]) and explained by phase field simulations (e.g. Ref. [25] using thin-film – i.e. stress-free – boundary conditions). Rather, because these vortices are demonstrated not to be present in the original specimen (i.e. prior to the lamella preparation needed to produce an electron-transparent sample), the point of novelty is that such specimen preparation process is therefore a route to producing topologically non-trivial polarization configurations, with potentially significant practical applications. I am not able to judge to what extent film thickness may implicitly have been appreciated to be a governing factor in earlier work on these materials, and the present manuscript does not explore where the transition region between the two endpoint regimes might be. Nevertheless, the case is clearly made, so if this insight is indeed novel then I think the manuscript would be of wide significance and interest and thus very suitable for publication in Nature Communications.

Important point for clarification:

* The lamella preparation in the methods section gives $\sim 30 \mu\text{m}$ as the thickness after mechanical polishing. Presumably this is not the specimen thickness for STEM imaging – the analysis of Fig. 2(d) especially would be questionable for a sample of thickness much beyond a few tens of nanometers. Since the evidence presented only demonstrated the presence of arrays of polarization vortices in the area of sample on which the EM data is obtained, can the authors please give a more concrete thickness estimate for the region(s) imaged in Fig. 2?

Minor points for the authors' consideration:

* The caption to Fig. 1(a) refers to a “yellow arrow line”, but all the arrows I can see are black. (Also on Fig. 1(a), the significance of the blue obelisk-shaped graphic element in the bottom left eluded me for a while, but from Supplementary Fig. 2 I tentatively guess it to denote the PFM tip. May I assume this is a standard convention in the PFM literature for denoting scan orientation?)

* It is asserted in line 131 of the manuscript that the interface between the PTO and STO layers is “sharp and coherent”. The latter I agree with. The former is relative. I'd not expect sharpness to be evident at the magnification at which Fig. 2(a) is shown. Supplementary Fig. 5 is more convincing, but it seems to me that in the STO layer at the bottom of that image the Sr map shows a weak Sr signal in the first Pb plane in the PTO layer above, and likewise that the Pb map shows a weak Pb signal in the first Sr plane in several of the STO layers adjacent to the PTO layers. This has little bearing on the

thrust of the authors' argument, but may warrant adding a little nuance to the statements about "atomically sharp"?

* With a convergence semi-angle of 30 mrad, the quoted collection inner angle of 39 mrad seems to me uncharacteristically low to qualify being called "high-angle" ADF. If that inner angle is not simply a typographical error, its relatively low value may complicate the interpretation of Fig. 2(d) because there may be an appreciable contribution from elastically-scattered electrons (although since the analysis presented is qualitative rather than quantitative it is quite difficult to imagine imaging artefacts giving rise to such clear topological features in the displacement maps). Can the authors offer any qualification or comment on this point?

* The manuscript largely reads very well, but the second sentence of Supplementary Fig. 2 is a bit off: "The scanangle will be seted to be 90°". Consider making "scan angle" two words, and replacing "will be seted" with "was set"?

Engineering polar vortex from topologically trivial domain architecture

Response to referees' comments

We would like to thank all the three referees for their thoughtful and highly constructive reviews that help us to improve our manuscript further. In response to their comments and suggestions, we have carried out additional experiments and analysis, with which we have revised our manuscript carefully to address all the concerns made by the referees. These revisions are detailed in our point-to-point responses below, and key revisions are also highlighted in the revised manuscript.

Referee #1

Comment #1

In this study, the superlattice that the authors claimed is $(\text{PbTiO}_3)_{10}/(\text{SrTiO}_3)_{10}$, however, according to the EDS maps shown in Supplementary Fig. 5, the superlattice is likely $(\text{PbTiO}_3)_{10-12}/(\text{SrTiO}_3)_9$ rather than $(\text{PbTiO}_3)_{10}/(\text{SrTiO}_3)_{10}$. Since the phase components of PTO/STO superlattice samples are very sensitive to layer thicknesses, the authors should accurately describe the geometry of the superlattice.

Response

We appreciate the reviewer's careful examination. We agree with the reviewer that the slight variation in thickness does exist in our sample, for example as shown in Fig. R1, though the typical thickness is $(\text{PbTiO}_3)_{10}/(\text{SrTiO}_3)_{10}$. We have clarified this point on page 5 along with additional data (Fig. R1 as Supplementary Fig. 7) for thickness variation:

“Higher magnification ADF-STEM image in Supplementary Fig. 7 shows slight thickness variation in PTO layers at the unit-cell scale, though the typical configuration is determined as $(\text{PTO})_{10}/(\text{STO})_{10}$.”

Fig. R1 ADF-STEM image of STO₁₀/PTO₁₀ superlattice and the corresponding line profile of the area labeled by the white short dash line, presenting the thickness of PTO and STO layers in term of the number of unit cells (u.c.).

Comment #2

It is known that DSO substrate features anisotropy and usually it has significant effects on the domain structures of ferroelectric films grown on it, I suggest the authors make some discussions on how the vortex phase looks like if the sample is cut from the perpendicular direction.

Response

We appreciate this important suggestion and we have carried out additional experiments to verify it. As shown in Fig. R2, cross-sectional DF-TEM of the same STO₁₀/PTO₁₀ superlattice sample taken along the orthogonal [100]_{pc} zone axis also

shows periodic contrast modulation along $[010]_{pc}$ direction, suggesting the presence of periodic clockwise-anticlockwise vortex pairs in PTO layers¹. The corresponding electron diffraction pattern contains satellite spots, further supporting the existence of periodic vortex array². We have added this point on page 6 along with additional data (Fig. R2 as Supplementary Fig. 8):

“To examine the effect of substrate anisotropy, thin lamella was also cut from superlattice film in a perpendicular direction, and the resulted cross-sectional DF-TEM (supplementary Fig. 8) taken along the orthogonal $[100]_{pc}$ zone axis also shows periodic contrast modulation along $[010]_{pc}$ direction. The corresponding electron diffraction pattern contains satellite spots as well, indicating the presence of periodic vortex arrays independent of lamella orientation with respect to the substrate. It is thus established that arrays of polar vortex have been engineered in PTO_{10}/STO_{10} lamellas from topologically trivial domain architecture in PTO_{10}/STO_{10} superlattice film.”

Fig. R2 a Cross-sectional DF-TEM image taken along the $[100]_{pc}$ (i.e. $[\bar{1}10]_o$) zone axis under two-beam condition by selecting 002_{pc} g-vector, with 20 nm scale bar. **b** The corresponding SAED pattern of (a) with enlarged $(001)_{pc}$ spots in the inset.

Comment #3

In Page 6, the authors describe that “.....the in-plane lattice parameter a and out-of-plane lattice parameter c are calculated to be $\sim 3.93 \text{ \AA}$ and $\sim 4.02 \text{ \AA}$ ”. I suggest that such a statement should be rearranged, since electron diffraction cannot provide such a high precision for determination of lattice parameters.

Response

We appreciate this important point, and we have revised the statement on page 5 as “It reveals superlattice reflections both in-plane and out-of-plane, from which the in-plane lattice parameter a and out-of-plane lattice parameter c are calculated to be $\sim 3.9 \text{ \AA}$ and $\sim 4.0 \text{ \AA}$, similar to those observed in reference 40⁴⁰”, given the precision of the selected area electron diffraction in TEM.

Comment # 4

Some of the description in the text should be tuned down, such as “From application point of view, it is highly desirable if we can engineer nontrivial polar topologies from the ordinary ferroelectrics with topological trivial domain architecture, yet this has not been accomplished to our best knowledge”. Actually, observation of a_1/a_2 to flux-closure domain transformations induced by e-beam was reported recently (Acta Materialia 193 (2020) 311).

Response

We appreciate this important reference, and we have added the corresponding discussion on page 2 and 3 with this reference:

“Recent study has observed the coexistence of flux-closure and a_1/a_2 strip-like domain in tensile-strained PbTiO_3 sandwiched between GdScO_3 substrate and a SrTiO_3 layer, while their transformation under electron beam illumination has also been demonstrated²⁴. These works raise the prospect of engineering topologically nontrivial polar topologies from topologically trivial domain architecture, which we seek to accomplish.”

And

“From an application point of view, it is highly desirable if we can engineer nontrivial polar topologies from the ordinary ferroelectrics with topologically trivial domain architecture, and with better understanding on energetics of polar topologies^{12,22}, we are now in a position to explore such possibility.”

Comment #5

Finally, I have to point out that the references cited in this manuscript particularly in the abstract and Introduction sections are randomly showed up in order, and consequently the contextualization of this work is not good enough. When taking about topological ferroelectric domains, one cannot start from the Nature (2016) work (Reference No.1 in the present manuscript) while ignoring the study that was published one year before (Y.L. Tang et al. Science 2015). One can see that the sample system in the Nature (2016) paper is the same as that in the Science paper (2015), and the domain configurations are similar in these two papers. Overviewing experimental observation on topological ferroelectric domains, relevant studies should be discussed in a chronological order, such as Science (2015) on flux-closure, Nature (2016) on vortices, Nature (2019) on skyrmion, and Nature Materials (2020) on meron. I find, in a very recent publication by the present collaborative team (Nature Communications 2020), they presented a very nice review on the recent progress of topological domains. I suggest the authors should maintain a single style on reviewing the same topic. In the high-profile journals like nature communications, a well-presented contextualization on the previous studies in the field is necessary -- a good review on literature is a kind of contribution as well to the academic community.

Response:

We thank the reviewer for the valuable comment, and we apologize for the oversight. We have revised this part as suggested, overviewing experimental observation on topological ferroelectric domains in a chronological order.

Referee #2

General Comment

As my first impression, the present manuscript could have been submitted to Nature as a Comment on "Observation of polar vortices in oxide superlattices." by Yadav, A. K. et al. Nature 530, 198-201 (2016) rather than an independent publication. It appears, however, that the authors chose the latter option. They wrote a manuscript

claiming that the reconstruction is useful as an artificial engineering technique of topologically non-trivial polar vortices from topologically trivial stripe domain, instead of criticizing the previous publication. I would like to leave the editor to choose the best way to publish their findings. Overall, their manuscript is well written based on technically sound experimental data with scientific rigor. Of course, the contents would attract a majority of professional and general readers as was proved by the previous Nature publication.

Response

We appreciate the referee's assessment of this work, and we choose to publish this finding as an independent paper instead of as a comment, which we feel more comfortable.

Major Revision

Comment #1

A plane-view DF-TEM image as shown in Fig. 3b in Ref. 1 must be presented. The data is significant to discuss how the two orthogonal thin plate fabrication processes reconstruct the original stripe domain.

Response:

We appreciate this important point and have carried out additional experiments. We acquired the plane-view DF-TEM image (Figure R3) of the PTO₁₀/STO₁₀ superlattices, from which the period of the strip-like domain is estimated to be ~69 nm, in good agreement with the PFM and RSM results in Figure 1f and 3f. In combination with the cross-sectional DF-TEM image taken along the orthogonal [100]_{pc} zone axis, as shown in Figure R1, the reconstruction into topologically non-trivial polar vortex from topologically trivial stripe domains is confirmed. We have added this point on page 7 with additional data (Fig. R3 as Supplementary Fig. 10) of plan-view DF-TEM image:

“We also acquired the planar-view DF-TEM image of the PTO₁₀/STO₁₀ superlattice film as shown in Supplementary Fig. 10, which exhibits long-range strip-like domain

along $[110]_{pc}$ direction with the period estimated to be ~ 69 nm consistent with both PFM and 3D-RSM data. These set of data thus reconfirms the topologically trivial strip-like domain structure in PTO_{10}/STO_{10} superlattice before reconstruction.”

Figure R3 **a** Planar-view STEM image of a PTO_{10}/STO_{10} superlattice film and the EDS mapping corresponding to the marked yellow box showing uniform distribution of elements. **b** Thickness profile corresponding to the blue arrow in (a) extracted from the STEM EELS data, showing that the thickness for the orange box in (a) ranges from ~ 150 to ~ 180 nm. **c** The planar-view DF-TEM imaging for orange box area in (a) exhibits long-range ordering along $[110]_{pc}$ direction and confirms the periodic a_1/a_2 strip-like domain in the superlattice films.

Comment #2

PFM measurements reveal the period of stripe domain is 68.2 ± 2.6 nm, which is much different from the period of vortices (8.6 nm) as well as their phase-field simulations (also 8.6 nm?). The authors must explain the discrepancy in the main text. In addition,

the orientation of the stripe domain (a-c domain) is different from the stripe (a-a domain) as shown in Fig. 3b and Fig. 4 in the Ref. 1. The authors must explain the difference in the main text as well.

Response:

We apologize for not making it clear that PFM maps strip domains before reconstruction, with a period of 68.2 ± 2.6 nm in good agreement with the ~ 70 nm periodicity determined from 3D-RSM measurement. After reconstruction, the period for vortex array is determined from DF-TEM to be ~ 8.6 nm, so PFM data also support our conclusion of reconstruction and there is no discrepancy.

Furthermore, as demonstrated in Figure 1 and Figure 3f, the stripe domain in the as-grown PTO₁₀/STO₁₀ superlattice is ~ 70 nm-wide planar a-a domain (instead of a-c domain) with the orientation of domain walls along $[1\bar{1}0]_{pc}$ direction, which is also demonstrated by the planar-view DF-TEM imaging in Figure R2. The stripe domain in Fig. 3b and Fig. 4 of the reference revealed by planar-view TEM is regarded as vortex array by the authors there, which we believe is due to different thickness of PTO layer, PTO₁₀/STO₁₀ versus PTO₁₆/STO₁₆.

We clarified these points on page 4 that:

“Domain widths for both a_1 and a_2 domains can be estimated from the LPFM mapping over a larger area in Supplementary Fig. 3 along the line profile shown, as presented in Fig. 1f, from which the period of the strip-like domain is estimated to be 68.2 ± 2.6 nm. Such domain pattern is quite common in ferroelectrics, for example in strained $Pb_xSr_{1-x}TiO_3$ ³⁵ and $PbTiO_3$ thin films³⁶, yet it is different from vortex array seen in PTO₁₆/STO₁₆ superlattice⁸, which we believe arises from different thickness. It will be exciting, however, if we can engineer such topologically trivial domain architecture into more interesting topologies, such as polar vortex”

And reemphasize the reconstruction on page 7 that:

“We also acquired the planar-view DF-TEM image of the PTO₁₀/STO₁₀ superlattice film as shown in Supplementary Fig. 10, which exhibits long-range strip-like domain

along $[110]_{pc}$ direction with the period estimated to be ~ 69 nm consistent with both PFM and 3D-RSM data. These set of data thus reconfirms the topologically trivial strip-like domain structure in PTO_{10}/STO_{10} superlattice before reconstruction.”

Comment #3

Introduction is too short (only 225 words). Please refer to Abid, A. Y. et al. Creating polar antivortex in $PbTiO_3/SrTiO_3$ superlattice. Nature Communications **12**, 2054, doi:10.1038/s41467-021-22356-0 (2021). A typical Introduction is composed of three paragraphs. It would be a good idea to include this latest publication in Introduction as it appears closely related with the present work.

Response

We appreciate this important point. As suggested, we have rewritten the Introduction section, which now has two paragraphs, and we added the latest publication on vortex-antivortex pair as the reference.

Comment #4

Also, Discussion is too short (only 122 words). It would be a good idea to add a discussion on the difference between the plan-view lamella and the cross-sectional lamella. The readers must be interested in how the two engineering processes reconstruct the original stripe domain.

Response

We appreciate this important point. As suggested, we have rewritten the Discussion part. In particular, we added discussion on page 8 regarding planar lamella:

“Such understanding can guide us exploring as well as engineering new polar topologies further. For example, we can explore planar instead of cross-sectional lamella that may modify the magnitude of biaxial misfit strain and thus change the energy landscape, which is currently under investigation.”

Minor Revisions

Comment #1

Throughout the manuscript: "topological trivial "-> "topologically trivial"

Response

We have corrected as suggested.

Comment #2

Title: "Reconstruction of polar vortex from topological trivial domain architecture"->

What is reconstructed is not the polar vortex but the topologically trivial domain.

"Reconstruction (or Engineering) of topologically trivial stripe domain into topologically non-trivial polar vortex " would be better.

Response:

Thank the reviewer for the valuable comment. We have revised the title as “**Engineering polar vortex from topological trivial domain architecture**”.

Comments

(3) Abstract, line 31: "Topological polar structures"->"Topologically non-trivial polar structures"

(4) Abstract, line 35: "artificially engineering nanostructured polar topologies"-> "artificially engineered polar topologies" as vortex, skyrmion, and meron possess non-trivial topology.

(5) Abstract, line 36: "reconstruction of topological trivial strip-like domain architecture commonly observed into arrays of polar vortex"->"reconstruction of topologically trivial strip-like domain architecture into arrays of polar vortex "

(6) Line 40: "polarization mapping⁷, atomic imaging⁸, and three-dimensional (3D) structure visualization⁹"->"polarization mapping by atomic imaging^{7,8} and threedimensional (3D) reciprocal space mapping⁹"

(7) Line 43: "tipping"-> The meaning is vague. Use a more comprehensive phrase.

Between line 46 and 47: Add "Introduction".

(8) Line 58: "From application point of view"->"From an application point of view"

- (9) Line 76: Insert “Results”.
- (10) Line 76: "Guided by previous report on strip-like domain configuration"-> A citation to the "previous report" is necessary.
- (11) Figure 2, caption, line 112: "scanning transmission electron microscopy (STEM) image"->"ADF-STEM image or ADF-STEM image", All acronyms must be defined in the main text when they appear first.
- (12) Figure 2, caption, line 113: "Dark field TEM image"->"DF-TEM image", All acronyms must be defined once in the main text when they appear first.
- (13) Figure 2, caption, line 115: "A selected-area electron diffraction (SAED) pattern"->"SAED pattern", All acronyms must be defined once in the main text when they appear first.
- (14) Line 136: “dark field TEM”->”dark field TEM (DF-TEM)”
- (15) Line 155: “Fig. 3f”->”Fig. 2f”, “Fig. 3e”->”Fig. 2e” (?)
- (16) Line 157: “dark field TEM”->” DF-TEM”
- (17) Line 239: “Our lamella configuration overcome such difficulty by accommodating arrays of zero-dimensional polar vortex, as exhibited by Fig. 4d-f.”-> Add more explanation on the usefulness of their engineering technique if possible.

Response

We have made all these correction as suggested.

Referee #3

Important point for clarification

The lamella preparation in the methods section gives ~30 μm as the thickness after mechanical polishing. Presumably this is not the specimen thickness for STEM imaging – the analysis of Fig. 2(d) especially would be questionable for a sample of thickness much beyond a few tens of nanometers. Since the evidence presented only demonstrated the presence of arrays of polarization vortices in the area of sample on which the TEM data is obtained, can the authors please give a more concrete thickness estimate for the region(s) imaged in Fig. 2?

Response: Thanks for the reviewer’s valuable comment and suggestion. As suggested, we have prepared additional lamella samples cut from superlattice thin film via focused ion beam (FIB), and acquired the corresponding DF-TEM images. The lamella thickness is estimated using STEM electron energy loss spectroscopy (EELS) method³⁻⁵, as shown in Figure R4. In particular, STEM-EELS was performed using an aberration-corrected FEI Titan Themis G2 at 300 KV. The typical energy resolution (half-width of the full zero-loss peak, ZLP) was 0.8 eV. For the different regions of superlattice lamella, STEM-EELS images in the low loss region were acquired and converted to a thickness map of inelastic mean free paths (t/λ) in Gatan’s Digital Micrograph software using the absolute log-ratio method for the zero-loss and the first plasmon peaks⁴. The thickness of the superlattice thin lamella with arrays of clockwise-anticlockwise vortices were quantitatively calculated to be in the range of ~20 - 40 nm as shown in Figure 4R b. We have added the details in Methods, and added the discussion on page 7 with additional data (Fig. R4 as Supplementary Fig. 5) on lamella thickness:

“For the following transmission electron microscopy (TEM) studies, we focused on a region of lamella with thickness in the range of ~20 - 40 nm determined using electron energy loss spectroscopy (EELS) method³⁷, as detailed in Supplementary Fig. 5.”

Figure R4 | **a** The lamella sample cut from superlattice film. **b** The DF-TEM image acquired from the blue box in (a). The upper left inset displays the STEM EELS thickness profile calculated using the log ratio method from a low-loss spectrum

image. Line profile extracted at the white arrow showing that the thickness ranges from ~20 to ~40 nm.

Minor points for the authors' consideration:

Comment #1

The caption to Fig. 1(a) refers to a “yellow arrow line”, but all the arrows I can see are black. (Also on Fig. 1(a), the significance of the blue obelisk-shaped graphic element in the bottom left eluded me for a while, but from Supplementary Fig. 2 I tentatively guess it to denote the PFM tip. May I assume this is a standard convention in the PFM literature for denoting scan orientation?)

Response:

Thank the reviewer to point out these issues. We have revised and added the related descriptions in the caption of Fig. 1 as “The **blue obelisk-shaped marker denotes the orientation of PFM cantilever. c, d** Amplitude and phase profiles along the **black arrow line** in (a, b).”. Furthermore, the cartoon-like tip denoting scan orientation is commonly adopted in PFM literature^{6,7}.

Comment #2

It is asserted in line 131 of the manuscript that the interface between the PTO and STO layers is “sharp and coherent”. The latter I agree with. The former is relative. I’d not expect sharpness to be evident at the magnification at which Fig. 2(a) is shown. Supplementary Fig. 5 is more convincing, but it seems to me that in the STO layer at the bottom of that image the Sr map shows a weak Sr signal in the first Pb plane in the PTO layer above, and likewise that the Pb map shows a weak Pb signal in the first Sr plane in several of the STO layers adjacent to the PTO layers. This has little bearing on the thrust of the authors’ argument, but may warrant adding a little nuance to the statements about “atomically sharp”?

Response:

We appreciate this point and revised it on page 5 as “**relatively sharp** and coherent”.

Comment #3

With a convergence semi-angle of 30 mrad, the quoted collection inner angle of 39 mrad seems to me uncharacteristically low to qualify being called “high-angle” ADF.

If that inner angle is not simply a typographical error, its relatively low value may complicate the interpretation of Fig. 2(d) because there may be an appreciable contribution from elastically-scattered electrons (although since the analysis presented is qualitative rather than quantitative it is quite difficult to imagine imaging artefacts giving rise to such clear topological features in the displacement maps). Can the authors offer any qualification or comment on this point?

Response

Thank the reviewer for the valuable comment. To increase the contrast of the vortex array structure, we acquired the ADF-STEM images with a smaller convergence semi-angle for enhancing the stress contrast in the strained PTO layer. Different convergence semi-angle will only affect the relative intensity of the atomic column, not the position of the atom columns. So, it does not affect our conclusion. To avoid confusion, we changed HAADF to ADF in the revised manuscript and added the related descriptions in methods on page 10.

Comment #4

The manuscript largely reads very well, but the second sentence of Supplementary Fig. 2 is a bit off: “The scanangle will be seted to be 90°”. Consider making “scan angle” two words, and replacing “will be seted” with “was set”?

Response

Thank the reviewer for the valuable comment and suggestion. We have revised this.

References

1. Yadav, A. K. *et al.* Observation of polar vortices in oxide superlattices. *Nature* **530**, 198-201 (2016).
2. Chen, P. *et al.* Atomic imaging of mechanically induced topological transition of ferroelectric vortices. *Nat. Commun.* **11**, 1840 (2020).
3. Conlan, A. P., Tillotson, E., Rakowski, A., Cooper, D. & Haigh, S. J. Direct measurement of TEM lamella thickness in FIB-SEM. *Journal of Microscopy* **279**,

- 168-176 (2020).
4. Egerton, R. F. Electron energy-loss spectroscopy in the electron microscope. *Springer*, 293-397 (2011).
 5. Egerton, R. F. Electron energy-loss spectroscopy in the TEM. *Rep. Prog. Phys.* **72**, 016502 (2008).
 6. Ma, J. *et al.* Controllable conductive readout in self-assembled, topologically confined ferroelectric domain walls. *Nat. Nanotechnol.* **13**, 947-952 (2018).
 7. Kim, K.-E. *et al.* Configurable topological textures in strain graded ferroelectric nanoplates. *Nat. Commun.* **9**, 403 (2018).

REVIEWERS' COMMENTS

Reviewer #1 (Remarks to the Author):

I find the authors have well addressed the concerns that I raised when reading the initial submission. In the meanwhile, I also find the responses by the authors to the concerns raised by the other reviewers are acceptable. Therefore, I am happy to recommend the paper for publication in Nature Communications.

X.L.Ma

Review of revised Nature Communications manuscript
"Engineering polar vortex from topologically trivial domain architecture" by
Congbing Tan et al. Manuscript ID: NCOMMS-21-11782A

I am pleased to find that their one-by-one responses to my previous comments are almost satisfactory. However, I would like to ask the authors to make the last revisions, which I believe are significant to deepen their findings. I would recommend the manuscript for a publication in *Nature Communications* after they respond to the major and minor revisions as described in the following paragraphs.

Major revision

(1) A plan-view DF-TEM image as shown in Supplementary Fig. 10c has to be replaced by a new image having the same magnification and orientation with that of Fig. 3b as shown in Ref. 8. See Fig. R1 below, which shows a comparison between the two images. In my opinion, it appears that a narrower periodicity than 69 nm (indicated by red arrows in Fig. R1b) exists. I hope a higher magnification image could deepen their understanding of the reconstruction (engineering). Please add a comment on the discrepancy of the two images in Discussion.

Fig. R1 a: Fig. 3b of Ref.8 with a scale bar of 100 nm, b: Supplementary Fig. 10c (rotated) with a scale bar of 200 nm, showing a narrower periodicity than 69 nm (indicated by red arrows).

Minor Revisions

- (1) Line 38: "Topologically nontrivial polar structures are not only attractive for high-density data storage, but may also enable ultralow power microelectronics thanks to their exotic negative capacitance" -> "Topologically nontrivial polar structures are attractive not only for high-density data storage but also for ultralow power microelectronics thanks to their exotic negative capacitance"
- (2) Line 48: "tipping the subtle electrostatic and elastostatic energetics and providing the driving force for the polar vortex formation" -> The meaning of "tipping" is vague for non-native English readers. Use a more plain word.
- (3) Line 65: PbTiO₃ -> PbTiO₃(PTO)

(4) Line 65: $\text{SrTiO}_3 \rightarrow \text{SrTiO}_3(\text{STO})$

(5) Line 81: $(\text{PbTiO}_3)_{10}/(\text{SrTiO}_3)_{10}(\text{PTO}_{10}/\text{STO}_{10}) \rightarrow (\text{PTO})_{10}/(\text{STO})_{10}$, Use the same expression throughout the following paragraphs.

End of Review

Reviewer #3 (Remarks to the Author):

The authors have addressed the points I raised.

I reiterate that the electron microscopy analysis seems to me to be thorough in showing that multiple forms of evidence point to a common conclusion, and that if the insight that controlling cross-sectional lamella thickness is a route to producing topologically non-trivial polarization configurations in such materials is indeed novel (something I am not sufficiently familiar with this class of materials to judge) then I highly recommend the manuscript for publication in Nature Communications.

Engineering polar vortex from topologically trivial domain architecture

Response to referees' comments

We would like to thank all the three referees for their thoughtful reviews and positive recommendations. For the remaining issues raised by the second referee, we have carried out additional experiments and analysis, with which we have revised our manuscript accordingly. These revisions are detailed in our point-to-point responses below, and key revisions are also highlighted in the revised manuscript.

Referee #1

Comment

I find the authors have well addressed the concerns that I raised when reading the initial submission. In the meanwhile, I also find the responses by the authors to the concerns raised by the other reviewers are acceptable. Therefore, I am happy to recommend the paper for publication in *Nature Communications*.

Response

We appreciate the positive recommendation of the referee.

Referee #2

Comment #1

I am pleased to find that their one-by-one responses to my previous comments are almost satisfactory. However, I would like to ask the authors to make the last revisions, which I believe are significant to deepen their findings. I would recommend the manuscript for a publication in *Nature Communications* after they respond to the major and minor revisions as described in the following paragraphs.

Major revision

A plan-view DF-TEM image as shown in Supplementary Fig. 10c has to be replaced by a new image having the same magnification and orientation with that of Fig. 3b as

shown in Ref. 8. See Fig. R1 below, which shows a comparison between the two images. In my opinion, it appears that a narrower periodicity than 69 nm (indicated by red arrows in Fig. R1b) exists. I hope a higher magnification image could deepen their understanding of the reconstruction (engineering). Please add a comment on the discrepancy of the two images in Discussion.

Response:

We appreciate this important suggestion and we have acquired additional plane-view DF-TEM images with different magnifications, as shown in Figure R1. As the reviewer pointed out, there exist a few strip-like domains with narrower periodicity (indicated by red arrows), and there are a small number of strip-like domains with perpendicular orientation as well, which can also be observed by PFM over a larger area (Figure R2). The orientations of these domain walls are fully consistent with continuum analysis based on strain and polarization compatibilities, and it is distinct from the orientations and period of vortex array reported in Ref. 8 (Figure R3). In other words, these are still topologically trivial domain pattern.

To clarify this point, we have updated Supplementary Figure 10 (combining DF-TEM and PFM) as Figure R4, and added the following discussions on page 7:

“We also acquired the planar-view DF-TEM image of the (PTO)₁₀/(STO)₁₀ superlattice film as shown in Supplementary Fig. 10, which exhibits long-range strip-like domain along $[110]_{pc}$ direction with the period estimated to be ~ 69 nm

consistent with both PFM and 3D-RSM data. There exist a few strip-like domains with narrower periodicity, and there are a small number of strip-like domains with perpendicular orientation along $[-110]$ as well (Supplementary Fig. 10d), which can also be observed by PFM over a larger area (Supplementary Fig. 10e). These topologically trivial domain patterns are fully consistent with continuum analysis based on strain and polarization compatibilities⁴, and it is distinct from the orientations and period of vortex array reported in Ref. 8. These set of data thus reconfirms the topologically trivial strip-like domain structure in $(\text{PTO})_{10}/(\text{STO})_{10}$ superlattice before reconstruction.”

Fig. R1 a-c Different-magnification planar-view DF-TEM imaging of a $(\text{PbTiO}_3)_{10}/(\text{SrTiO}_3)_{10}$ superlattice exhibits long-range ordering along $[110]_{\text{pc}}$ direction and confirms the periodic a_1/a_2 strip-like domain in the superlattice films. It also exists narrower-periodicity strip-like domains (indicated by red arrows).

Fig. R2 **a** In-plane(IP) PFM amplitude image and **b** IP-PFM phase image of $5 \times 5 \mu\text{m}^2$ area of the as-grown $(\text{PTO})_{10}/(\text{STO})_{10}$ superlattice shows a few stripe-like domain structure in perpendicular direction in local regions.

Fig. R3 **a** The planar-view DF-TEM imaging of a $(\text{PbTiO}_3)_{16}/(\text{SrTiO}_3)_{16}$ superlattice copied from Fig. 3b of Ref.8 (Yadav et al. Nature 530, 198 (2016)). **b** The planar-view DF-TEM imaging of a $(\text{PbTiO}_3)_{10}/(\text{SrTiO}_3)_{10}$ superlattice exhibits long-range ordering along $[110]_{\text{pc}}$ direction and confirms the periodic a_1/a_2 strip-like domain in the superlattice films. The scale bar is 100 nm.

Fig. R4 a A planar-view STEM image of a $(\text{PTO})_{10}/(\text{STO})_{10}$ superlattice film and the EDS mapping corresponding to the marked yellow box showing uniform distribution of elements of PTO/STO superlattice film and the DyScO_3 substrate. **b** Thickness profile corresponding to the blue arrow in (a) extracted from the STEM EELS data, showing that the thickness for the orange box in (a) ranges from ~ 150 to ~ 180 nm (denoted by red line), exceeding the thickness of the $(\text{PTO})_{10}/(\text{STO})_{10}$ superlattice films. **c** The planar-view DF-TEM imaging for orange box area in (a) exhibits long-range in-plane ordering along $[110]_{\text{pc}}$ direction and confirms the periodic a_1/a_2 strip-like domain in the superlattice films. **d** The local-enlarged planar-view DF-TEM imaging in (c) shows a few stripe-like domain structure in perpendicular direction. **e** IP-PFM amplitude image of $5 \times 5 \mu\text{m}^2$ area of the as-grown $(\text{PTO})_{10}/(\text{STO})_{10}$ superlattice shows that there are a small number of strip-like domains with perpendicular orientation as well, in good agreement with the DF-TEM results in (c) and (d).

Comment #2

Minor Revisions

(1) Line 38: "Topologically nontrivial polar structures are not only attractive for high-density data storage, but may also enable ultralow power microelectronics thanks to their exotic negative capacitance"->"Topologically nontrivial polar structures are attractive not only for high-density data storage but also for ultralow power microelectronics thanks to their exotic negative capacitance"

(2) Line 48: "tipping the subtle electrostatic and elastostatic energetics and providing the driving force for the polar vortex formation"-> The meaning of "tipping" is vague for non-native English readers. Use a more plain word.

(3) Line 65: PbTiO_3 -> $\text{PbTiO}_3(\text{PTO})$

(4) Line 65: SrTiO_3 -> $\text{SrTiO}_3(\text{STO})$

(5) Line 81: $(\text{PbTiO}_3)_{10}/(\text{SrTiO}_3)_{10}(\text{PTO}_{10}/\text{STO}_{10})$ -> $(\text{PTO})_{10}/(\text{STO})_{10}$, Use the same expression throughout the following paragraphs.

Response

We appreciate these important suggestions and we have made all these correction as suggested.

Referee #3

Comment

The authors have addressed the points I raised.

I reiterate that the electron microscopy analysis seems to me to be thorough in showing that multiple forms of evidence point to a common conclusion, and that if the insight that controlling cross-sectional lamella thickness is a route to producing topologically non-trivial polarization configurations in such materials is indeed novel (something I am not sufficiently familiar with this class of materials to judge) then I highly recommend the manuscript for publication in Nature Communications.

Response

We appreciate the positive recommendation of the referee.